# Atherogenesis in *Apoe*^−/−^ and *Ldlr*^−/−^ Mice with a Genetically Resistant Background

**DOI:** 10.3390/cells12091255

**Published:** 2023-04-26

**Authors:** Hideyuki Torikai, Mei-Hua Chen, Li Jin, Jiang He, John F. Angle, Weibin Shi

**Affiliations:** 1Department of Radiology and Medical Imaging, University of Virginia, Charlottesville, VA 22908, USA; 2Orthopedic Surgery, University of Virginia, Charlottesville, VA 22908, USA

**Keywords:** atherosclerosis, hyperlipidemia, genetic background, small dense LDL, carotid artery, oxidative stress

## Abstract

*Apoe*-deficient (*Apoe*^−/−^) and *Ldlr*-deficient (*Ldlr*^−/−^) mice are two common animal models of hypercholesterolemia and atherosclerosis. The two models differ in lipid and glucose metabolism and other mechanisms involved in atherogenesis. Here we examined atherosclerotic lesion formation in the two models with an atherosclerosis-resistant C3H/HeJ (C3H) background. 3-month-old C3H-*Ldlr*^−/−^ and C3H-*Apoe*^−/−^ mice developed minimal atherosclerotic lesions in the aortic root when fed a chow diet. After 12 weeks on a Western diet, C3H-*Ldlr*^−/−^ mice developed 3-fold larger lesions than C3H-*Apoe*^−/−^ mice in the aortic root (127,386 ± 13,439 vs. 41,542 ± 5075 μm^2^/section; *p* = 0.00028), but neither knockout formed any lesion in the carotid artery. After being ligated near its bifurcation, the common carotid artery developed intimal lesions in both knockouts 4 weeks after ligation, significantly larger in C3H-*Ldlr*^−/−^ than C3H-*Apoe*^−/−^ mice (68,721 ± 2706 vs. 47,472 ± 8146 μm^2^/section; *p* = 0.028). Compared to C3H-*Apoe*^−/−^ mice, C3H-*Ldlr*^−/−^ mice showed a 50% reduction in plasma MCP-1 levels, similar levels of malondialdehyde, an oxidative stress biomarker, on both chow and Western diets, but higher small dense LDL levels on the Western diet. These results suggest a more significant role for small dense LDL than inflammation and oxidative stress in the different susceptibility of the mouse models to atherosclerosis.

## 1. Introduction

Atherosclerosis is the pathological cause of coronary artery disease and ischemic stroke, which are the leading causes of death globally [1]. Dyslipidemia, featured by elevations in plasma LDL cholesterol and triglyceride levels and reductions in HDL cholesterol levels, is a major risk factor for atherosclerosis. Prospective population and cohort studies have associated total, LDL, HDL cholesterol, and triglyceride with the risk of cardiovascular events [2,3]. A recent large-scale genome-wide association study (GWAS) of coronary artery disease, which includes a quarter of a million cases combined from existing GWAS studies with data from the Million Veteran Program, identified 95 novel loci significantly associated with the disease, and 36 of them harbor the genes affecting blood lipid levels or hyperlipidemia [4]. The effectiveness of lipid-lowering therapies with statins and anti-PCSK9 antibodies in reducing cardiovascular events has provided definitive evidence for a causal role of dyslipidemia in atherosclerosis [5,6]. Even among middle-aged individuals without conventional cardiovascular risk factors, LDL cholesterol levels show an independent association with atherosclerosis presence and extent [7].

*Apoe*-deficient (*Apoe*^−/−^) and *Ldlr*-deficient (*Ldlr*^−/−^) mice are two extensively used animal models of dyslipidemia and atherosclerosis. On a chow diet, *Apoe*^−/−^ mice have an elevated plasma cholesterol level mainly from elevations in VLDL and chylomicron remnants, and *Ldlr*^−/−^ mice have an increased cholesterol level due to an accumulation of LDL [8,9]. When fed a Western diet, *Ldlr*^−/−^ mice develop as severe hypercholesterolemia as *Apoe*^−/−^ mice [10,11]. The two models also differ in glucose metabolism and other phenotypes that are associated with atherogenesis [8,12]. Development of atherosclerosis has been compared with the two models on an atherosclerosis-susceptible C57BL/6 (B6) background [9,13]. When fed the Paigen diet containing high fat, high cholesterol, and cholate for 1, 2, or 3 months, *Apoe*^−/−^ mice had higher plasma cholesterol levels at the first two time points and a comparable cholesterol level at the last time point and developed larger aortic lesions than *Ldlr*^−/−^ mice across all 3 time points [13]. Increased atherosclerotic lesion formation of *Apoe*^−/−^ mice relative to *Ldlr*^−/−^ mice has also been observed when fed a Western diet [14]. However, after being genetically modified to express ApoB100, *Ldlr*^−/−^ mice exhibited a 3-fold increase in aortic lesion sizes relative to *Apoe*^−/−^ mice despite their comparable plasma cholesterol levels [9]. Higher numbers of small dense ApoB100–containing lipoproteins were considered responsible for increased atherosclerosis in *Ldlr*^−/−^ mice. Moreover, both Ldlr and ApoE are expressed in vascular, immune, and other types of cells, and thus, they may also contribute to atherosclerosis through mechanisms independent of hypercholesterolemia [12].

The genetic background has a dramatic influence on the effect of *Apoe* and other genes on atherosclerosis in mice. Indeed, *Apoe*^−/−^ mice with a B6 background develop advanced atherosclerotic lesions in the carotid artery, those with a C3H background form no atherosclerosis in the artery, and those with a BALB/c background form a lesion in between [15,16]. The insensitivity of arterial wall cells to oxidized LDL-induced inflammatory gene expression is partially responsible for the resistance of C3H mice to atherosclerosis [17]. In this study, we examined atherosclerotic lesion formation in *Ldlr*^−/−^ mice with the C3H background through comparison with C3H-*Apoe*^−/−^ mice and explored likely mechanisms underlying their differential atherosclerotic lesion formation.

## 2. Materials and Methods

### 2.1. Mice

C3H-*Ldlr*^−/−^ mice and C3H-*Apoe*^−/−^ mice at N10 or more backcrossed generations were generated in our laboratory using the classical congenic breeding strategy [10]. To study primary atherosclerosis, the two knockouts of mice were kept on a regular chow or fed a Western diet containing 21% fat, 48.5% carbohydrate, 17% protein, and 0.2% cholesterol by weight (TD 88137, Envigo, Indianapolis, IN, USA). Mice were started on the Western diet at 6 weeks of age and kept on the diet for 12 weeks. To study lesion formation in the ligated carotid artery, mice were switched onto the Western diet one week before surgery and remained on the diet thereafter. All procedures were performed according to an animal protocol approved by the Institutional Animal Care and Use Committee (animal protocol #: 3109).

### 2.2. Ligation and Morphometric Analysis of Common Carotid Arteries

The procedure for ligating the left common carotid artery was performed as described [16]. Four weeks after surgery, mice were euthanized via prolonged inhalation of isoflurane. The vasculature was perfused first with saline and then with 10% formalin via the heart. The entire neck was cut and fixed in 10% formalin for >48 h. The front soft tissues of the neck encompassing the left and right common carotid arteries were dissected out, embedded in a freezing medium (Tissue-Tek, Sakura Finetek USA, Torrance, CA, USA) and cross-sectioned in 10-µm thickness. Serial sections were mounted on slides with 10 sections per slide. Three evenly spaced slides were stained with oil red O and hematoxylin and counterstained with fast green. Morphometric measurements of common carotid arteries were made on one section per stained slide using Zeiss Zen software, including the circumferences of the lumen and internal and external elastic laminae on digital images. The intimal lesion area was calculated by subtracting the luminal area from the area encircled by the internal elastic lamina, and the medial area was calculated by subtracting the area encircled by the internal elastic lamina from the area defined by the external elastic lamina. Measurements made from 3 separate slides were averaged for each vessel, and this average was used for statistical analysis.

### 2.3. Measurement of Primary Atherosclerosis

Atherosclerotic lesion sizes in the aortic root and the left carotid artery bifurcation of mice that were fed 12 weeks of a Western diet were measured as reported [15,16]. Briefly, the vasculature was perfusion-fixed with 10% formalin. The aortic root and adjacent heart, the distal left common carotid artery and adjacent branches were dissected out, embedded, and cross-sectioned in 10-μm thickness. Sections were stained with oil red O and hematoxylin and counterstained with fast green. Atherosclerotic lesion areas were measured using Zeiss AxioVision 4.8 software. Results were expressed as the average of lesion areas on 5 sections with the largest readings for each mouse.

### 2.4. Immunohistochemical Analysis

The presence of macrophages and smooth muscle cells in atherosclerotic lesions was detected by immunostaining using the avidin-biotinylated peroxidase system (Vecter Laboratories, Newark, CA, USA), as described [18,19]. Primary antibodies used include rat anti-mouse α-smooth muscle actin IgG (ARG63621, Arigo Biolaboratories, Burlington, ON, Canada) and rat anti-mouse Mac-3 IgG1 (Clone M3/84, BD Biosciences, San Jose, CA, USA).

### 2.5. Measurement of Malondialdehyde and MCP-1

Plasma MCP-1 was measured with an enzyme-linked immunosorbent assay (ELISA) kit by following the manufacturer’s instructions (R&D Systems, Minneapolis, MN, USA). Plasma levels of malondialdehyde, a product of lipid peroxidation, were measured with Cayman Thiobarbituric Acid Reactive Substances (TBARS) kit (Cat. # 10009055).

### 2.6. Small Dense LDL Assay

Plasma levels of small dense LDL were measured using a modified two-step method previously reported [20]. In the first step, non-small dense LDL, IDL, VLDL lipoproteins, and chylomicrons were precipitated by incubating plasma with a phosphotungstate-magnesium salt precipitating reagent (FUJIFILM Wako Diagnostics, Mountain View, CA, USA), pH 10, in a ratio of 1:1 (chow diet) or 1:2 (Western diet) for 15 min at room temperature. After centrifugation at 15,000 rpm for 10 min, the supernatant was collected. In the second step, ApoB concentration in the supernatant was determined by ELISA (MyBiosource, San Diego, CA, USA; Cat. #: MBS937790). The effectiveness of this method in the preparation of small dense LDL was validated by cryogenic transmission electron microscopy. The pellet was suspended with phosphate-buffered saline (PBS). To be representative, 3 individual pellet or supernatant samples were pooled in an equal proportion. An aliquot of the sample (~3.5 μL) was applied on a glow-discharged, perforated carbon-coated grid (2/1-3C C-Flat; Protochips, Raleigh, NC, USA), manually blotted with filter paper, and rapidly plunged into liquid ethane. Low-dose images were collected on a Tecnai F20 Twin transmission electron microscope (FEI, Hillsboro, OR, USA). As shown in Figure 1, particles in the supernatant were small, <10 nm. In contrast, the sizes of particles in the pellet varied, with most particles having a larger diameter.

In addition, Sampson’s equation was used to estimate large buoyant and small dense LDL cholesterol concentrations: Large buoyant LDL cholesterol was calculated as 1.43 × LDL − [0.14 × In (triglyceride) × LDL] − 8.99, and small dense LDL was calculated as the difference between total LDL and large buoyant LDL cholesterol levels [21].

### 2.7. Statistical Analysis

Results were expressed as means ± SE, with “n” indicating the number of animals. Student’s *t*-test was used to determine the statistical significance of differences between two groups in various measurements. Differences were considered statistically significant at *p* ≤ 0.05.

## 3. Results

### 3.1. Primary Atherosclerosis

At 3 months of age, on a chow diet, C3H-*Ldlr*^−/−^ mice and C3H-*Apoe*^−/−^ mice developed minimal atherosclerotic lesions in the aortic root. The average lesion size of C3H-*Ldlr*^−/−^ mice was larger than the lesion size of C3H-*Apoe*^−/−^ mice (1820 ± 1473 vs. 879 ± 763 µm^2^/section), though the difference was not statistically significant (*p* = 0.17) (Figure 2). After being fed the Western diet for 12 weeks, C3H-*Ldlr*^−/−^ mice exhibited a 3-fold increase in aortic lesion size relative to that of C3H-*Apoe*^−/−^ mice (127,386 ± 38,012 vs. 41,542 ± 16,048 µm^2^/section; *p* = 0.0002).

Atherosclerotic lesions of both knockouts stained intensely red with oil red O (Figure 3). Immunostaining showed an abundance of macrophages in atherosclerotic lesions. Smooth muscle staining was observed in the cap, the underneath lesion, and the medial arterial wall. An overlying thin fibrous cap was observable in atherosclerotic lesions of C3H-*Apoe*^−/−^ mice. In contrast, there was no concrete fibrous cap, although numerous smooth muscle cells were present at the top portion of plaques in C3H-*Ldlr*^−/−^ mice.

Atherosclerotic lesion formation in the carotid arteries was examined for C3H-*Ldlr*^−/−^ and C3H-*Apoe*^−/−^ mice that were fed 12 weeks of the Western diet. Neither knockout developed any atherosclerotic lesion in the arteries. Areas encircled by the external and internal elastic laminae near carotid bifurcation were smaller in C3H-*Ldlr*^−/−^ than C3H-*Apoe*^−/−^ mice (51,501 ± 14,376 vs. 80,197 ± 21,331 μm^2^; 22,095 ± 10,201 vs. 39,087 ± 17,625 μm^2^, respectively), although the differences were not statistically significant (*p* > 0.05) (Figure 4). The medial area of carotid arteries was also smaller in C3H-*Ldlr*^−/−^ mice (29,406 ± 5938 vs. 41,111 ± 6407 μm^2^; *p* = 0.21).

### 3.2. Lesion Formation in Ligated Carotid Artery

Morphometric measurements were made on ligated left common carotid artery and contralateral right common carotid artery for *Apoe*^−/−^ and *Ldlr*^−/−^ mice of both sexes 4 weeks after ligation. *Ldlr*^−/−^ mice had an intimal lesion size of 68,721 ± 2706 μm^2^/section in ligated arteries, significantly larger than 47,472 ± 8146 μm^2^/section of *Apoe*^−/−^ mice (*p* = 0.029) (Figure 5). The medial area of ligated arteries was smaller in *Ldlr*^−/−^ mice than that of *Apoe*^−/−^ mice (21,397 ± 1313 vs. 27,800 ± 3631 μm^2^/section), though the difference did not reach statistical significance (*p* = 0.12). The areas encircled by the internal (IEL) and external elastic laminae (EEL) were comparable between *Ldlr*^−/−^ and *Apoe*^−/−^ mice (IEL: 77,076 ± 3050 vs. 70,673 ± 7327 μm^2^/section; EEL: 98,473 ± 3370 vs. 98,473 ± 9171 μm^2^/section; *p* > 0.05).

Compared to contralateral common carotid arteries, ligated arteries of both knockouts showed significant increases in the area encircled by either the external or internal elastic lamina (Figure 5). The cross-sectional area of the medial layer of ligated arteries was significantly increased in *Apoe*^−/−^ mice (27,800 ± 3631 vs. 18,973 ± 1180 μm^2^/section; *p* = 0.038) but not in *Ldlr*^−/−^ mice (21,397 ± 1313 vs. 18,894 ± 941 μm^2^/section; *p* = 0.15).

### 3.3. Plasma MCP-1 Concentration

Plasma MCP-1 levels were measured for mice of both sexes under both feeding conditions. *Ldlr*^−/−^ mice had significantly lower MCP-1 levels than *Apoe*^−/−^ mice on either chow (39.8 ± 9.4 vs. 73.7 ± 7.6 pg/mL; *p* = 0.023) or Western diet (84.8 ± 17.2 vs. 141.8 ± 13.9 pg/mL; *p* = 0.017) (Figure 6). Compared to the chow diet, the Western diet significantly raised plasma MCP-1 levels in both *Ldlr*^−/−^ mice (*p* = 0.036) and *Apoe*^−/−^ mice (*p* = 0.00055).

### 3.4. Oxidative Stress Biomarker

Plasma levels of malondialdehyde, a product of lipid peroxidation, were determined using TBARS assay. *Ldlr*^−/−^ and *Apoe*^−/−^ mice of both sexes showed no significant differences in plasma malondialdehyde levels on either chow (25.9 ± 2.9 vs. 24.1 ± 2.9 µM) or Western diet (58.6 ± 5.4 and 50.2 ± 7.4 µM) (Figure 7). Compared to the chow diet, the Western diet significantly elevated plasma malondialdehyde levels in both *Ldlr*^−/−^ mice (*p* = 0.0018) and *Apoe*^−/−^ mice (*p* = 0.022).

### 3.5. Plasma Small Dense LDL

Plasma small dense LDL was determined by measuring ApoB concentration in the supernatant after precipitation of other ApoB-containing lipoproteins. On a chow diet, *Ldlr*^−/−^ mice of both sexes had a lower ApoB level than *Apoe*^−/−^ mice (428 ± 30 vs. 480 ± 51 µg/mL), although the difference was not statistically significant (*p* = 0.39) (Figure 8). On the Western diet, ApoB level was significantly higher in *Ldlr*^−/−^ mice than *Apoe*^−/−^ mice (836 ± 43 vs. 678 ± 38 µg/mL; *p* = 0.017). Compared to the chow diet, the Western diet significantly elevated small dense LDL levels of both *Ldlr*^−/−^ mice (*p* = 6.5 × 10^−8^) and *Apoe*^−/−^ mice (*p* = 0.008) based on ApoB amounts.

Small dense LDL cholesterol concentrations in *Ldlr*^−/−^ and *Apoe*^−/−^ mice were also estimated from standard lipid panel results using the Sampson equation [21]. On the chow diet, female *Apoe*^−/−^ mice had higher small dense LDL cholesterol levels than *Ldlr*^−/−^ counterpart (70.0 ± 3.9 vs. 55.2 ± 5.9 mg/dL; *p* = 0.041) (Figure 8). On the Western diet, female *Ldlr*^−/−^ mice had higher small dense LDL cholesterol levels than *Apoe*^−/−^ mice (227.1 ± 13.4 vs. 190.5 ± 26.4), though the difference was not statistically significant (*p* = 0.25). The Western diet significantly raised small dense LDL cholesterol levels of both knockouts (*p* < 0.05). The same trend was seen in male mice (Appendix A).

## 4. Discussion

In this study, we compared *Ldlr*^−/−^ mice with *Apoe*^−/−^ mice for the development of atherosclerosis when they were on the atherosclerosis-resistant C3H background. The two knockouts developed minimal lesions in the aortic root on a chow diet, but C3H-*Ldlr*^−/−^ mice developed 3-fold as large lesions as C3H-*Apoe*^−/−^ mice after 12 weeks on the Western diet. Both knockouts developed no lesions in the carotid artery even after 12 weeks on the Western diet, but C3H-*Ldlr*^−/−^ mice developed significantly larger intimal lesions than C3H-*Apoe*^−/−^ mice in the common carotid artery after blood flow was interrupted. C3H-*Ldlr*^−/−^ mice had 50% lower plasma MCP-1 levels and similar malondialdehyde levels on both chow and Western diets but higher small dense LDL levels on the Western diet compared to C3H-*Apoe*^−/−^ mice.

As seen in B6-*Ldlr*^−/−^ mice [22,23], chow-fed C3H-*Ldlr*^−/−^ mice developed atherosclerotic lesions in the aortic root despite the much smaller size. Unlike B6-*Ldlr*^−/−^ mice, which develop atherosclerotic lesions in the brachiocephalic artery when fed a chow diet [22], C3H-*Ldlr*^−/−^ mice developed no atherosclerotic lesions in the carotid artery even after 12 weeks of the Western diet. C3H-*Apoe*^−/−^ mice also develop no atherosclerotic lesions in the carotid artery on the Western diet [16]. However, once blood flow was interrupted by ligation at its distal end, the common carotid artery of both *Ldlr*^−/−^ and *Apoe*^−/−^ mice developed intimal lesions on the Western diet, which were significantly larger in *Ldlr*^−/−^ mice. Carotid ligation resulted in blood flow cessation in the common carotid artery, though the artery still experienced blood pressure and arterial pulsation. The interruption to the blood flow leads to the rapid development of endothelial dysfunction and focal inflammation, which are key elements contributing to the initiation and progression of atherosclerosis [15,19,24]. Neutrophil adhesion to the endothelium of ligated arteries is observable one day after ligation [19]. Increased endothelial expression of ICAM-1 and VCAM-1 was observed one-day post-ligation, and endothelial dysfunction was seen seven days post-ligation [24].

In this study, we found that C3H-*Ldlr*^−/−^ mice developed significantly larger atherosclerotic lesions in the aortic root than C3H-*Apoe*^−/−^ mice on the Western diet. This finding is opposite to what was observed when the two knockouts were on the B6 background [13]. An explanation for the discrepant results is the relative magnitude of hypercholesterolemia between the two knockouts. With the B6 background, *Apoe*^−/−^ mice had higher serum cholesterol levels than *Ldlr*^−/−^ mice during most of the 3-month high-fat feeding period [13]. In contrast, C3H-*Apoe*^−/−^ mice had a plasma cholesterol level comparable to that of C3H-*Ldlr*^−/−^ mice after 3 months of the Western diet [8]. Moreover, the two knockouts with the B6 background were fed a cholate-containing high-fat diet, while the mice in this study were fed a Western diet containing no cholate. The cholate diet induces more severe inflammatory responses in the liver and probably other tissues of B6 mice relative to C3H mice [25], and *Apoe*^−/−^ mice have shown an increased response to inflammatory stimuli such as dextran sodium sulfate compared to *Ldlr*^−/−^ mice [26]. Inflammation is an important mechanism underlying the development of atherosclerosis. In this study, we observed that *Ldlr*^−/−^ mice had significantly lower plasma MCP-1 levels than *Apoe*^−/−^ mice under either chow or Western diet, suggesting a lower inflammatory status. Decreased inflammatory responses in terms of Tnf-α, IL-1β, and IL-6 induction in the colorectum were observed in *Ldlr*^−/−^ mice relative to *Apoe*^−/−^ mice following treatment with azoxymethane and dextran sodium sulfate [26]. Besides its significant role in lipoprotein clearance by serving as a high-affinity ligand for both LDLR and LDLR-related proteins, ApoE has anti-inflammatory, antiproliferative, and anti-oxidative effects that suppress the development of atherosclerosis [12]. It attenuates unresolvable inflammation by binding to activated complement C1q protein [27]. As previously reported [11], we found that the Western diet dramatically elevated plasma MCP-1 levels in mice. MCP-1 is a proinflammatory cytokine produced by macrophages, endothelial cells, and smooth muscle cells and is involved in macrophage recruitment during atherogenesis [28]. Obviously, the decreased inflammatory state could not explain the increased atherosclerosis of *Ldlr*^−/−^ mice relative to *Apoe*^−/−^ mice. ApoE exerts an antiproliferative effect on smooth muscle cells [29], a major cellular component in advanced atherosclerotic plaques.

Hyperlipidemia and hyperglycemia increase the production of reactive oxygen species in vivo [30], which oxidize lipids and lipoproteins. Malondialdehyde is a secondary product resulting from the peroxidation of polyunsaturated fatty acids and represents a sensitive biomarker of oxidative stress [31]. In this study, we found that *Ldlr*^−/−^ and *Apoe*^−/−^ mice had similar plasma levels of malondialdehyde on either diet. Thus, oxidative stress could not explain the difference between the two knockouts in atherosclerosis susceptibility. As the levels of malondialdehyde were markedly elevated by feeding the Western diet, oxidative stress should contribute to increased plaque formation of the two knockouts on the high-fat diet.

On the chow diet, C3H-*Ldlr*^−/−^ mice had a total plasma cholesterol level of ~300 mg/dL and glucose level of 120 mg/dL, significantly lower than the total cholesterol level of 400 mg/dL and glucose level of 220 mg/dL in *Apoe*^−/−^ mice while on the Western diet, the two knockouts had similar total plasma cholesterol (>900 mg/dL) and glucose levels (300 mg/dL) [8]. These data were obtained from the same animals used in the current study, suggesting that variation in plasma cholesterol and glucose levels cannot explain the differential susceptibility of the two knockouts to atherosclerosis on the Western diet.

Small dense LDL particles are considered to have a higher atherogenic potential than large buoyant LDL particles. They can more easily penetrate into the arterial wall, where they become oxidized and subsequently promote focal inflammation and foam cell formation, contain fewer antioxidants, and are, thus, more susceptible to oxidation than large LDL particles [32]. Here, we found that C3H-*Ldlr*^−/−^ mice had higher small dense LDL levels than C3H-*Apoe*^−/−^ mice, though their HDL, non-HDL cholesterol, and triglyceride were similar on the Western diet [8]. ApoB-containing lipoproteins, including LDL, are taken up and removed from circulation through the LDL receptor (LDLR) and the LDL receptor-related protein (LRP). LDLR binds with ApoB-100 for the removal of LDL through endocytosis, and this process is facilitated by ApoE [33]. LRP does not directly interact with ApoB but rather uses ApoE as its main apolipoprotein ligand [34]. *Ldlr* deficiency leads to elevations of ApoB-100-containing lipoproteins, particularly LDL [35], and *Apoe* deficiency results in elevations of VLDL and chylomicron remnants [36]. In humans, nondietary (hepatic) lipoproteins only contain ApoB-100, while in mice, nondietary lipoproteins include both ApoB-48 and ApoB-100. On the chow diet, C3H-*Apoe*^−/−^ mice had slightly higher small dense LDL levels than C3H-*Ldlr*^−/−^ mice, and this might be attributable to their >2-fold higher non-HDL cholesterol levels [8]. The Western diet, which raises non-HDL cholesterol levels, also elevated small dense LDL levels of both knockouts.

Wild-type mice, including C3H, show negative remodeling or shrinkage of the ligated carotid artery [37]. Here, both C3H-*Ldlr*^−/−^ and C3H-*Apoe*^−/−^ mice exhibited increases in the areas encircled by the internal or external elastic lamina of ligated arteries. Positive vascular remodeling has also been observed in ligated arteries of other *Apoe*^−/−^ mouse strains [15,19] and human coronary arteries with atherosclerosis [38]. Smooth muscle cells are the major cellular component of intimal lesions in the ligated carotid artery of wild-type mice [39], while the intimal lesions of hyperlipidemic *Apoe*^−/−^ and *Ldlr*^−/−^ mice contain macrophages and other types of leukocytes. Macrophages produce MMP-9, MMP-12, and other enzymes that degrade the extracellular matrix [40], weakening the arterial wall with plaques and enhancing blood pressure-induced outward remodeling, while smooth muscle cells and the extracellular matrix produced by the cells restrain vessel distension. A higher macrophage count and lipid content are associated with a positive remodeling in the coronary artery with atherosclerotic plaques [41,42].

## 5. Conclusions

We demonstrated that *Ldlr*^−/−^ mice are more susceptible to Western diet-accelerated atherosclerosis than *Apoe*^−/−^ mice on the atherosclerosis-resistant C3H genetic background. *Ldlr*^−/−^ mice had significantly higher small dense LDL, lower proinflammatory cytokine MCP-1, and similar malondialdehyde levels relative to *Apoe*^−/−^ mice. Thus, small dense LDL concentrations appeared to contribute more significantly to differential atherosclerosis susceptibility of the two knockouts than inflammation and oxidative stress. Nevertheless, this remains speculative until further study has proven the causal role of small dense LDL in the increased susceptibility of C3H-*Ldlr*^−/−^ mice to atherosclerosis.

## Figures and Tables

**Figure 1 cells-12-01255-f001:**
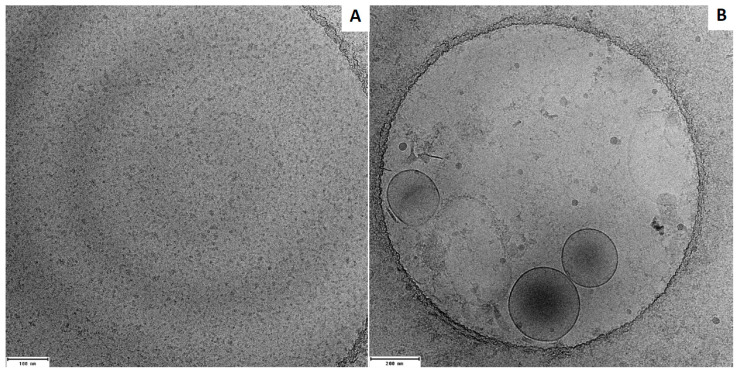
Electron microscopic images of lipoprotein particles in the supernatant (**A**) and pellet (**B**) after plasma was incubated with a precipitating reagent and then centrifuged. Note differences in particle sizes between the two samples.

**Figure 2 cells-12-01255-f002:**
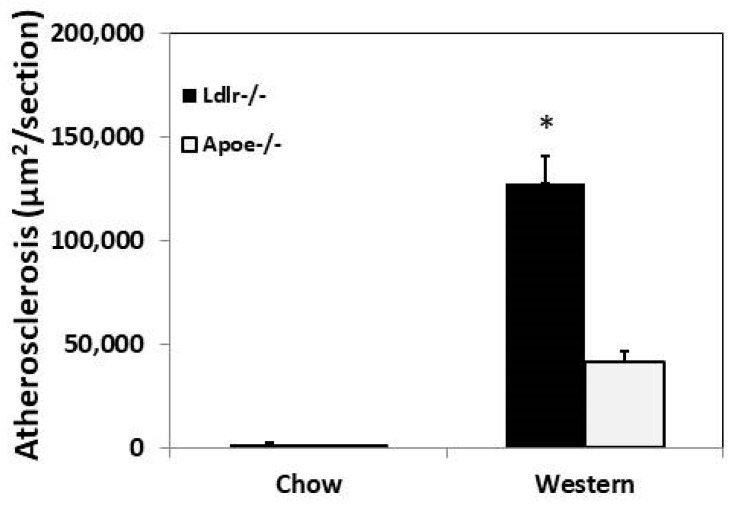
Atherosclerotic lesion sizes in the aortic root of female C3H-*Ldlr*^−/−^ mice and C3H-*Apoe*^−/−^ mice fed a chow or Western diet. Results are means ± SE of 6 to 10 mice group. * *p*  <  0.05 compared with C3H-*Apoe*^−/−^ mice on the same diet.

**Figure 3 cells-12-01255-f003:**
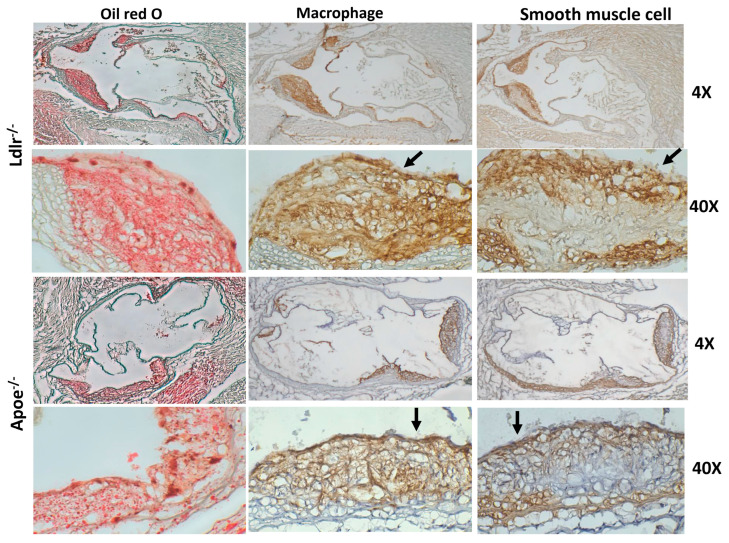
Representative micrographs showing histological findings in oil red O stained (**left** panel) and immunostained sections (**middle** and **right** panel) of the aortic root of C3H-*Ldlr*^−/−^ and C3H-*Apoe*^−/−^ mice fed a Western diet. Cryosections were stained with oil red O for neutral lipids and immunostained for macrophages and smooth muscle cells. Arrows point at the fibrous cap of atherosclerotic lesions. Note a thin overlying fibrous cap in *Apoe*^−/−^ and no concrete fibrous cap in the lesion of *Ldlr*^−/−^ mice. Original magnification ×4 (**top** row), ×40 (**bottom** row).

**Figure 4 cells-12-01255-f004:**
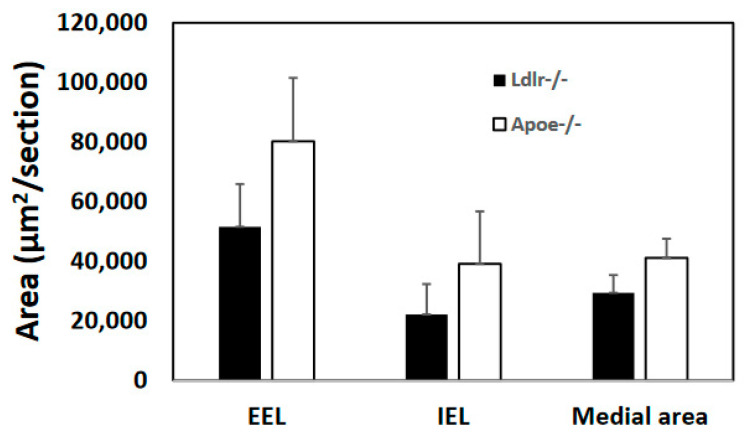
Morphometric measurement of areas encircled by the intimal and external elastic laminae in the left common carotid artery bifurcation of C3H-*Ldlr*^−/−^ and C3H-*Apoe*^−/−^ mice fed 12 weeks of the Western diet. Medial wall area was calculated as the difference between areas encircled by the external and internal elastic laminae. Values are means ± SE of 5 to 6 mice per group.

**Figure 5 cells-12-01255-f005:**
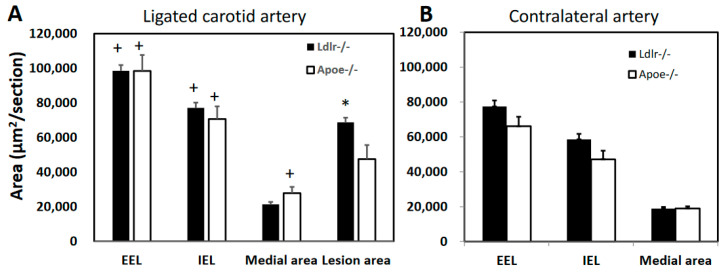
Morphometric assessments of ligated left common carotid artery (**A**) and contralateral right common carotid artery (**B**) of *Ldlr*^−/−^ and *Apoe*^−/−^ mice fed the Western diet. Measurements were made on carotid arteries 4 weeks after ligation. Values are means ± SE of 6 to 12 mice per group. * *p* < 0.05 versus *Apoe*^−/−^ mice, and + *p* < 0.05 versus contralateral carotid artery in the same group of mice.

**Figure 6 cells-12-01255-f006:**
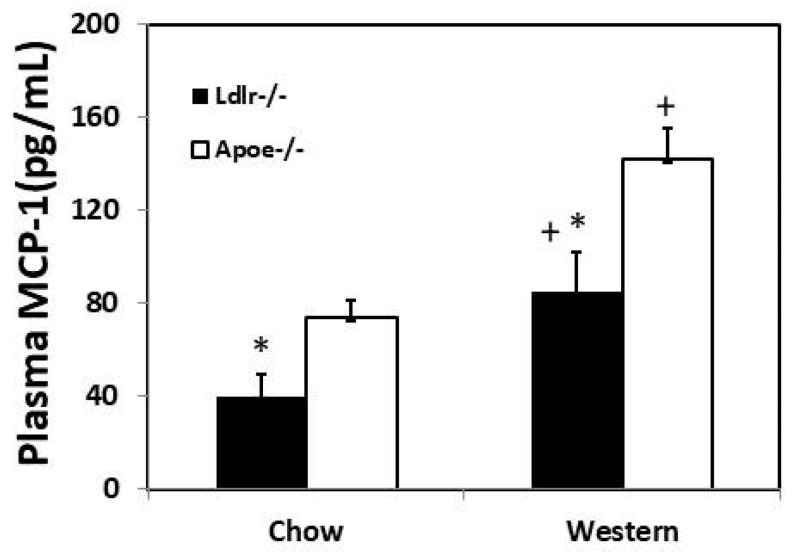
Plasma MCP-1 levels of *Ldlr*^−/−^ and *Apoe*^−/−^ mice fed a chow or Western diet. Values are means ± SE of 5 to 13 mice per group. * *p* < 0.05 versus *Apoe*^−/−^ mice, and + *p* < 0.05 versus chow diet.

**Figure 7 cells-12-01255-f007:**
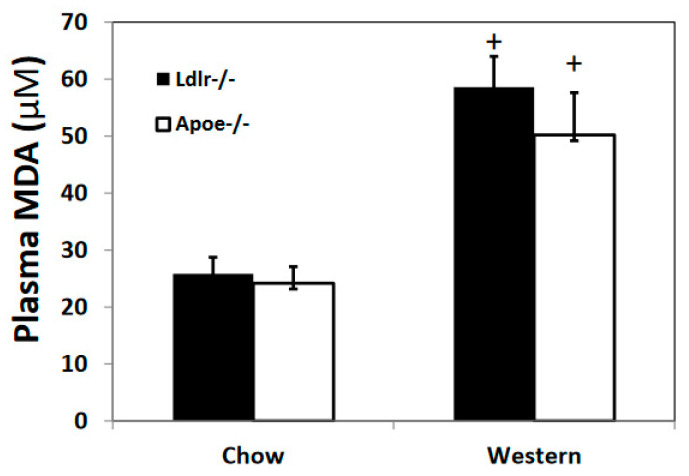
Plasma malondialdehyde levels of *Ldlr*^−/−^ and *Apoe*^−/−^ mice on both chow and Western diets. Values are means ± SE of 5 mice per group. + *p* < 0.05 versus chow diet.

**Figure 8 cells-12-01255-f008:**
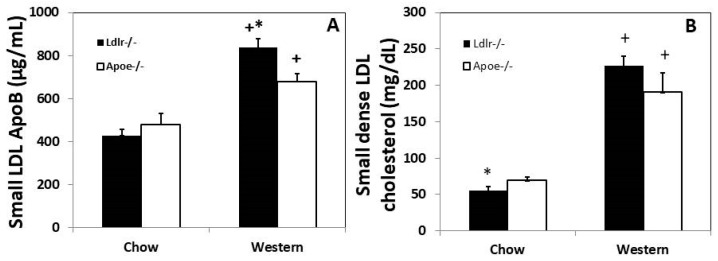
Plasma small dense LDL levels of *Ldlr*^−/−^ and *Apoe*^−/−^ mice on both chow and Western diets. (**A**), ApoB concentrations in plasma small dense LDL determined by ELISA. Results are means ± SE for 9 to 17 mice per group. (**B**), small dense LDL cholesterol levels estimated from standard lipid panel results using the Sampson equation. Results are means ± SE for 6 to 29 mice per group. * *p*  <  0.05 compared with *Apoe*^−/−^ mice, and ^+^
*p* < 0.05 versus chow diet.

## Data Availability

All data reported are included in Appendix A.

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
