# Peer review of "Atherogenesis in Apoe−/− and Ldlr−/− Mice with a Genetically Resistant Background"

_cells, 2023, doi:10.3390/cells12091255_

Round 1

Reviewer 1 Report

In this manuscript, authors found that Ldlr -/- mice are more susceptible to Western diet-accelerated atherosclerosis than Apoe-/- mice on a C3H genetic background. Small dense LDL level is higher and proinflammatory cytokine MCP-1 level was significantly lower in Ldlr-/- mice. LDL particle size and concentration might contribute to atherosclerosis than inflammation. 

However, before acceptation, there may be some corrections and clarities.

1. dyslipidemia induced by atherosclerotic bleeding, and then the effects of LDL on atherosclerotic bleeding, stratified laminating. However, it should be noted that APOE knockdown primarily affects the hepatic absorption of chylomicrons and hence the plasma cholesterol content, and that ADLR knockdown primarily affects the hepatic uptake and clearance of circulating LDL-critical proteins and hence the development of atherosclerosis in mice. 

2. The article mentions gene modifications that express APOB100 in row 55 but does not explain that APOB100 is an LDL receptor recognized carrier protein that is a key protein in liver uptake and elimination of circulating LDL. Increased APOB100 lipoprotein is a consequence of, but not a cause of, higher LDL content. 

3. The discussion part is the summary of this article. The language is rather refined, but there are places where the narrative is slightly confused and difficult to understand. 

4. The basic structure of the text involves multiple variables, sometimes interlacing narratives simultaneously, and is somewhat confusing. The references to small dense LDL, inflammation, and oxidative stress as the three factors in the main study comparisons are noted in the abstracts, but there are significantly fewer studies on oxidative stress than the first two. 

5. Blocking blood flow by ligation of a carotid artery branch has multiple effects on multiple sites, but it is unclear what factors are specifically controlled in this study. 

6.  The authors concluded that the two mice had almost identical levels of malondialdehyde (row 280), but in Figure 6 it can be observed that the mda levels of both mice increased by nearly 10 units after 12 weeks of a western diet, and the authors did not indicate here that there was no significant statistical difference, in the event that this difference led to a misinterpretation of the experimental results. 

7. In the introduction, I would add the concepts of atherosclerosis, apolipoprotein, and lipoprotein receptor (Ldlr), as well as the mechanism of action. 

8. I believe that the addition of cholesterol determination will make the results more accurate and convincing.

Details on references:

Reference 36 is repeated with reference 12. In ref. 12, the introduction section on apoe-/- and ldlr-/- mice uses the same words as this manuscript. The Discussion section summarizes the findings but leaves out the significance of the study, the outlook for the study, the problems in the trial, and the measures for improvement. 

Lline233, Ref. 12: No relevant Sampson content was found in the references. 

Lline270, Ref.29,30: The reference mentions the effect of a Western diet on a significant increase in VCAM, but not the effect of a Western diet on MCP-1 mentioned in that sentence. 

Lline261: Obviously, variation in plasma cholesterol and glucose levels cannot explain their different susceptibilities to atherosclerosis. This conclusion is based only on data from the literature, and these two values were not measured in this study.

Author Response

Comment 1. dyslipidemia induced by atherosclerotic bleeding, and then the effects of LDL on atherosclerotic bleeding, stratified laminating. However, it should be noted that APOE knockdown primarily affects the hepatic absorption of chylomicrons and hence the plasma cholesterol content, and that ADLR knockdown primarily affects the hepatic uptake and clearance of circulating LDL-critical proteins and hence the development of atherosclerosis in mice.

Response: Indeed, ApoE and LDLR affect lipoprotein clearance through different pathways as the reviewer stated.  The lipid profile of the two knockouts varies and is also affected by diet.  Apoe-/- mice develop more severe hypercholesterolemia than Ldlr-/- mice on a low fat chow, with the former being featured by accumulation of chylomicrons and VLDL remnants in the plasma and the latter mainly due to accumulation of LDL (https://www.ncbi.nlm.nih.gov/pmc/articles/PMC5001905/ ).  Apoe-/- mice also have lower HDL cholesterol levels compared to Ldlr−/− mice.  When fed a Western diet, C3H-Ldlr-/- and C3H-Apoe-/- mice have similar plasma levels of non-HDL, HDL cholesterol and triglyceride (https://www.ncbi.nlm.nih.gov/pmc/articles/PMC9220196/).  In this study, we have demonstrated that C3H-Ldlr-/- mice have higher small dense LDL particle levels than C3H-Apoe-/- mice, which may explain their increased susceptibility to atherosclerosis. 

Besides its lipoprotein clearance function, ApoE exerts anti-oxidative and anti-inflammatory effects that suppress the development of atherosclerosis (https://www.ncbi.nlm.nih.gov/pmc/articles/PMC5001905/).  Our present results that C3H-Apoe-/- mice have higher plasma MCP-1 levels and similar malondialdehyde levels compared to C3H-Ldlr-/- mice preclude the significance of oxidative stress and inflammation in the differential susceptibility to atherosclerosis.  We have now addressed the issue in Discussion.   

Comment 2. The article mentions gene modifications that express APOB100 in row 55 but does not explain that APOB100 is an LDL receptor recognized carrier protein that is a key protein in liver uptake and elimination of circulating LDL. Increased APOB100 lipoprotein is a consequence of, but not a cause of, higher LDL content.

Response: The article cited on transgenic expression of ApoB100 in Ldlr-/- and Apoe-/- mice supports the significance of LDL particle size relative to plasma cholesterol levels in modulation of atherosclerosis susceptibility.  We agree with the reviewer’s comment on the interaction between LDLR and ApoB100 that leads to LDL clearance.  Unlike human liver which produces ApoB-100 but not ApoB-48, mouse liver produces mostly ApoB-48 and a small fraction of ApoB-100 (https://www.sciencedirect.com/science/article/pii/S0022227520321738 ).  We have addressed this issue in Discussion.

Comment 3. The discussion part is the summary of this article. The language is rather refined, but there are places where the narrative is slightly confused and difficult to understand.

Response: The discussion part has been reworked to make it more easily understandable.

Comment 4. The basic structure of the text involves multiple variables, sometimes interlacing narratives simultaneously, and is somewhat confusing.  The references to small dense LDL, inflammation, and oxidative stress as the three factors in the main study comparisons are noted in the abstracts, but there are significantly fewer studies on oxidative stress than the first two.

Response: Oxidative stress intertwines with dyslipidemia, inflammation and endothelial dysfunction to affect the development and progression of atherosclerosis.  Comparable plasma levels of malondialdehyde, which is a sensitive biomarker of oxidative stress, suggest that oxidative stress is unlikely to contribute to the differential atherosclerosis susceptibility of the two knockouts.    

Comment 5. Blocking blood flow by ligation of a carotid artery branch has multiple effects on multiple sites, but it is unclear what factors are specifically controlled in this study.

Response: Carotid ligation results in rapid development of atherosclerosis in Apoe-/- and Ldlr-/- mice.  The major factors for this are the disturbed blood flow and ensuing endothelial dysfunction and inflammation.  Neutrophil adhesion to the endothelium of ligated arteries is observable one day after ligation (Oncotarget 2017;8:110289.  Increased adhesion molecule expression in partially ligated arteries was observed 1 day postligation (Am J Physiol Heart Circ Physiol 2009; 297: H1535).  Ligated carotid artery also shows rapidly developed endothelial dysfunction.  We have addressed this issue in Discussion.   

Comment 6.  The authors concluded that the two mice had almost identical levels of malondialdehyde (row 280), but in Figure 6 it can be observed that the mda levels of both mice increased by nearly 10 units after 12 weeks of a western diet, and the authors did not indicate here that there was no significant statistical difference, in the event that this difference led to a misinterpretation of the experimental results.

Response: Both knockouts show significantly elevations in plasma levels of malondialdehyde on the Western diet compared to the chow diet.  However, there are no significant differences between the two knockouts on either chow or Western diet.  We have addressed this issue in Discussion.  

Comment 7. In the introduction, I would add the concepts of atherosclerosis, apolipoprotein, and lipoprotein receptor (Ldlr), as well as the mechanism of action.

Response: The rationale for the study of C3H-Apoe-/- and Ldlr-/- mice has been reworded and justified, i.e., whether the two knockouts have differences in atherogenesis, if yes, why.    

Comment 8. I believe that the addition of cholesterol determination will make the results more accurate and convincing.

Response: Lipid profiles of these two knockouts with the C3H background have been recently reported (https://pubmed.ncbi.nlm.nih.gov/35740449/).  This study has forwarded the previous study by showing C3H-Ldlr-/- mice had higher small dense LDL levels than C3H-Apoe-/- mice on a Western diet. 

Details on references:

Comment Reference 36 is repeated with reference 12. In ref. 12, the introduction section on apoe-/- and ldlr-/- mice uses the same words as this manuscript. The Discussion section summarizes the findings but leaves out the significance of the study, the outlook for the study, the problems in the trial, and the measures for improvement.

Response: We thank the reviewer for finding the error. Ref. 36 has now been eliminated.

Comment Lline233, Ref. 12: No relevant Sampson content was found in the references.

Response: Revised.

Comment Lline270, Ref.29,30: The reference mentions the effect of a Western diet on a significant increase in VCAM, but not the effect of a Western diet on MCP-1 mentioned in that sentence.

Response: Revised.

Comment Lline261: Obviously, variation in plasma cholesterol and glucose levels cannot explain their different susceptibilities to atherosclerosis. This conclusion is based only on data from the literature, and these two values were not measured in this study.

Response: The data referred from the literature was achieved from the same animals used in the present study. Clarification has now been made.    

Reviewer 2 Report

Torikai et al report the development of atherosclerosis in Apoe-/- and Ldlr-/- mice with a genetic background of C3H/HeJ mice, which are resistant to atherosclerosis.  They found that both pieces of knockouts developed little atherosclerotic lesions in the aortic root on a chow diet.  When fed a Western diet, Ldlr-/- mice developed larger lesions than Apoe-/- mice in aortic root.  The two knockouts developed no lesion in the carotid artery, but after being ligated, Ldlr-/- mice formed larger lesions than Apoe-/- mice in the vessel.  Plasma levels of inflammatory and oxidative stress biomarkers and small dense LDL were compared between the two knockouts.  The increased atherosclerosis of Ldlr-/- mice was attributed to small dense LDL.  A lot of nice work has been done in the study and the findings are interesting. However, there are a couple of concerns that need to be addressed:

1)      With a B6 genetic background, Apoe-/- mice develop larger atherosclerosis than Ldlr-/- mice, opposite to the finding of this study. What your explanation for the discrepant results?

2)      Small dense LDLs are measured with a previously reported two-step method with modifications.  Have you validated your results with a different method?

Author Response

Comment 1: With a B6 genetic background, Apoe-/- mice develop larger atherosclerosis than Ldlr-/- mice, opposite to the finding of this study. What your explanation for the discrepant results?

Response: With the B6 background, Apoe-/- mice have higher serum cholesterol levels than Ldlr-/- mice during most of the 3-month high fat feeding period (https://www.ahajournals.org/doi/full/10.1161/01.ATV.16.8.1013).  In contrast, C3H-Apoe-/- mice have a plasma cholesterol level comparable to C3H-Ldlr-/- mice after 3 months of Western diet (https://www.mdpi.com/2227-9059/10/6/1429).  Also, the two knockouts with the B6 background were fed a cholate-containing high-fat diet, while the mice in this study were fed a Western diet containing no cholate.  Cholate diet induces inflammatory responses in the liver and probably other tissues in vivo, and Apoe-/- mice have shown an increased response to inflammatory stimuli such as dextran sodium sulfate compared to Ldlr-/- mice.  Inflammation is an important mechanism in the development of atherosclerosis.  Hypercholesterolemia is synergistic with inflammation to promote atherosclerosis in Apoe-/- mice.

Comment 2: Small dense LDLs are measured with a previously reported two-step method with modifications.  Have you validated your results with a different method?

Response: The effectiveness of the two-step method in measurement of small dense LDL has been validated with cryo-EM.  Representative EM images are shown in figure 1.

Round 2

Reviewer 1 Report

The manuscript has been well revised, but still can be better improved if make corrections as follows.

1. The grouping of mice at lines 74–77 of the original text is not clear enough: it does not specify the types of mice here, but only indicates the grouping of mice in terms of diet. They should be divided into four groups: APOE(-/-)mice chow diet,Ldlr(-/-) mice chow diet,APOE(-/-)mice "western food"diet and Ldlr(-/-)mice "western food"diet.

2. The horizontal headings of Figuer23 are chaotic. The first column is titled "oil red O," which indicates a staining method, while the second and third are titled "macrophage" and "smooth muscle cell", respectively, which indicate the structure name. The three headings should be unified as the staining method or the structure name.

3. The number of mice in each group is not uniform. For example, the average value of Figure 34 came from 5 to 6 mice, while that of Figure 54 came from 6 to 12 mice. Please explain. 

4. The gender of each group of mice was ambiguous. As shown in Figure 12, it was clearly stated that the mouse was female, but the mice mentioned later didn’t mention gender.

5. The conclusion that carotid artery ligation leads to accelerated initiation and progression of atherosclerosis (lines 272-280) by comparing left carotid artery ligation with right carotid artery without ligation mentioned in the study is not related to the conclusion drawn in this paper.

6. In Section 4 discuss at the end of the paper, the conclusion that small dense LDL contributes more to atherosclerosis than oxidative stress drawn through the analysis and the data mentioned above is illogical and lacks a theoretical basis and verification.

7. The final conclusion of this paper is seriously inconsistent with the title "Atherogenesis in Apoe-/- and Ldlr-/- mice with a genetically resistant background", and the correlation between genetic background and atherosclerosis in mice has not been mentioned.

8. There is an error in line 268 (double "even")

9. There are three phenomena after the ligation near the carotid bifurcation (including the words after "though") in lines 273-276. However, in the next sentence "this leads to", the specific meaning of "this" is not clear (guess the author refers to "the blood flow cessation in the common carotid artery"). If the above three consequences have the same effect, "this" should not be used.

10. "Post ligation" in line 279 should be post-ligation. 

Author Response

Comment 1. The grouping of mice at lines 74–77 of the original text is not clear enough: it does not specify the types of mice here, but only indicates the grouping of mice in terms of diet. They should be divided into four groups: APOE(-/-)mice chow diet,Ldlr(-/-) mice chow diet,APOE(-/-)mice "western food"diet and Ldlr(-/-)mice "western food"diet.

Response: Amended.

Comment 2. The horizontal headings of Figuer23 are chaotic. The first column is titled "oil red O," which indicates a staining method, while the second and third are titled "macrophage" and "smooth muscle cell", respectively, which indicate the structure name. The three headings should be unified as the staining method or the structure name.

Response: Revised to “histological findings in oil red O (left panel) and immunostained sections (middle and right panel) …”.

Comment 3. The number of mice in each group is not uniform. For example, the average value of Figure 34 came from 5 to 6 mice, while that of Figure 54 came from 6 to 12 mice. Please explain. 

Response: The two knockouts were generated in our lab.  These mice were hard to breed and their litter sizes were small.  It is noteworthy that both knockouts are inbred and genetically identical so individual phenotypic variation should be small.   

Comment 4. The gender of each group of mice was ambiguous. As shown in Figure 12, it was clearly stated that the mouse was female, but the mice mentioned later didn’t mention gender.

Response: Amended.

Comment 5. The conclusion that carotid artery ligation leads to accelerated initiation and progression of atherosclerosis (lines 272-280) by comparing left carotid artery ligation with right carotid artery without ligation mentioned in the study is not related to the conclusion drawn in this paper.

Response: We talked about the ligation model and likely mechanisms for enhanced atherogenesis.

Comment 6. In Section 4 discuss at the end of the paper, the conclusion that small dense LDL contributes more to atherosclerosis than oxidative stress drawn through the analysis and the data mentioned above is illogical and lacks a theoretical basis and verification.

Response: Revised.